# Patient-Oriented Research to Improve Internet-Delivered Cognitive Behavioural Therapy for People of Diverse Ethnocultural Groups in Routine Practice

**DOI:** 10.3390/healthcare11152135

**Published:** 2023-07-26

**Authors:** Ram P. Sapkota, Emma Valli, Andrew Wilhelms, Kelly Adlam, Lee Bourgeault, Vanessa Heron, Kathryn Dickerson, Marcie Nugent, Heather D. Hadjistavropoulos

**Affiliations:** Online Therapy Unit, Department of Psychology, University of Regina, 3737 Wascana Parkway, Regina, SK S4S 0A2, Canada; ram.sapkota@uregina.ca (R.P.S.); emma.valli@uregina.ca (E.V.); andrew.wilhelms@uregina.ca (A.W.); kelly.adlam@uregina.ca (K.A.); lee.bourgeault@uregina.ca (L.B.); vanessa.heron@uregina.ca (V.H.); kathryn.dickerson@uregina.ca (K.D.); marcie.nugent@uregina.ca (M.N.)

**Keywords:** internet-delivered therapy, depression, anxiety, cultural adaptation, patient-oriented research, digital health

## Abstract

There has been limited research on improving Internet-delivered Cognitive Behavioural Therapy (ICBT) in routine online therapy clinics that serve people from diverse ethnocultural groups (PDEGs). This article describes a patient-oriented adaptation approach used to address this gap in research. A working group consisting of people with lived experience, community representatives, ICBT clinicians, managers, and researchers was formed. The working group examined archival feedback on ICBT from past patients who self-identified as being from diverse ethnocultural backgrounds (N = 278) and the results of interviews with current patients (N = 16), community representatives (N = 6), and clinicians (N = 3). The archival data and interviews revealed the majority of the patients reported being satisfied with and benefitting from ICBT. Suggestions for improvement were not related to the cognitive-behavioural model and techniques, but rather to making treatment materials more inclusive. Consequently, the ICBT adaptation focused on adding content related to cultural influences on mental health, addressing stigma, diversifying case stories, examples, and imagery, adding audiovisual introductions, and replacing English idioms with more descriptive language. Moreover, further training was offered to clinicians, and efforts were made to improve community outreach. This study demonstrates a process for using patient-oriented research to improve ICBT within routine care serving patients of diverse backgrounds.

## 1. Introduction

Internet-delivered cognitive behaviour therapy (ICBT) represents a growing method of offering mental health care whereby treatment materials are shared online as lessons. This form of care is often offered with therapist support provided via secure messages or phone calls. There is a strong body of research showing that this approach is effective and can produce similar results to face-to-face therapy, e.g., [1,2], while also reducing common barriers to receiving mental health care related to location, limited time, and stigma [3]. It represents an important development as mental health problems are prevalent, disabling, and often undertreated due to barriers to care. Specifically, one in five Canadians is estimated to experience a mental illness in any given year, and one in three is likely to experience a mental illness during their lifetime [4]. Major depression and anxiety disorders affect approximately 5.4% and 4.6%, respectively, of the Canadian population [5]. Since the coronavirus disease of 2019 (COVID-19) pandemic, People of Diverse Ethnocultural Groups (PDEGs) (for the lack of better terminology, we have used the phrase People of Diverse Ethnocultural Groups (PDEGs) to refer to all but self-identified Caucasian/White people in this study. In this regard, PDEGs encompass Indigenous Canadians as well as “visible minorities” or “racialized minorities” such as black, Asian, and Latin American. Other terms for this group that are used in the literature include visible monitories, racialized minorities, or Black Indigenous People of Colour (BIPOC)) have reported higher rates of mental health concerns, e.g., [6,7,8].

Despite the prevalence of mental illness, a large number of Canadians do not seek and/or receive adequate psychological and mental health services for their problems [9,10]. A wide variety of barriers contribute to the undertreatment of mental health conditions, such as a shortage of psychological and psychiatric care practitioners and the consequent excessive wait times, lack of sufficient Federal as well as provincial funding to cover the cost of mental health services, language obstacles, and a lack of knowledge on where to access services [10,11]. Unfortunately, this leads to many Canadians enduring physical and mental disability and suffering, economic productivity losses, and an elevated risk of mortality by suicide due to unmet mental health care needs [9,10,12,13]. Underutilization of mental health care appears to be even greater among PDEGs compared to other Canadians, e.g., [13,14,15], and PDEGs often struggle to overcome cultural barriers (e.g., stigma, perceived lack of cultural fit, help sought within family), which leads to poor access to, and utilization of existing formal mental health services [16,17,18,19,20]. Moreover, due to differences in cultural backgrounds, level of acculturation, lived experiences, and place of dwelling (urban vs. rural), not all PDEGs in Canada have the same understanding of, access to, or the perceived need for the same type and quality of mental health care and services, e.g., see [19,21,22,23]. Furthermore, this lack of access and utilization of mental health services by PDEGs has been attributed to the limited availability of culturally informed services and service providers [11,18,24]. In terms of ICBT, despite the benefits of ICBT as another option for mental health care, it has its own potential additional barriers. It requires access to a computer/smart phone, internet service, and a basic knowledge of the use of these devices and services [25,26], and these issues may be more prominent among PDEGs.

### 1.1. Cultural Influence and the Need for Cultural Adaptation 

It is fairly established in cross-cultural research literature that social, cultural, and contextual factors influence every aspect of mental health and illness. These factors play an important role in the genesis (e.g., what gets defined as a problem), experience, expression of symptoms, meaning attributions, coping, help seeking, and treatment outcomes of mental illnesses, e.g., [27,28,29]. Recognising the important role that culture (e.g., shared traditions, worldviews, norms, beliefs, values, knowledge, behaviours, etc.) plays in mental health and illness, the Diagnostic and Statistical Manual of Mental Disorders, 5th edition DSM-5 [30] introduced “Cultural Concepts of Distress” (CCD) and acknowledged that mental health concerns are “locally shaped entities” and all forms of mental illnesses are influenced by culture [30]. Likewise, a report published by the Lancet Commissions on Culture and Health highlights “the systematic neglect of culture in health and health care is the single biggest barrier to the advancement of the highest standard of health worldwide” [31]. 

An extensive body of literature supports with nearly universal agreement that a culturally informed (Also referred to as: culturally competent, culturally sensitive, culturally responsive, culturally commensurate, culturally attuned, culturally congruent, culturally relevant, culture-centred, etc.) approach to mental health care and research is essential to avoid potential mistakes in assessment, diagnosis, treatment, and intervention program development and implementation, e.g., [18,21,32,33,34,35,36,37,38,39]. Though less has been reported on the cultural adaptations of internet-based psychological interventions, systematic reviews and meta-analytic evidence support that culturally adapted face-to-face and internet-based psychological interventions are superior in effectiveness when compared to non-adapted interventions for PDEGs, e.g., [40,41,42,43,44,45,46]. Furthermore, two meta-analyses found that greater treatment effectiveness was associated with a higher number of cultural adaptations made to various elements of the intervention [42,47].

Several theoretical guides with a range of recommended methods—e.g., ecological validity [48], surface vs. deep structure adaptation [49], heuristic framework [50], cultural accommodation model [51], formative method [52], integrated strategy [53], cultural treatment adaptation framework [54], framework with three cultural adaptation elements (cultural concepts of distress, treatment components, and treatment delivery) [55], etc.—have been published to facilitate cultural adaptation of evidence-based psychological interventions. See [56] for an overview of existing cultural adaptation frameworks. Although this scholarship provides important guidance in the cultural adaptation of the content and process of psychological interventions to a specific cultural group, none of it is readily implementable in multicultural settings [37,53,54,55,57]. 

The Bernal and Bonilla [48] ecological validity model has been widely used and has had a major influence on the subsequent development of the aforementioned guides [58]. This model recommends making adaptations in eight dimensions, or elements of psychological intervention. These include: (1) language (i.e., use of culturally appropriate and syntonic language, patients ’ native language); (2) persons (i.e., ethnocultural matching of patient and therapists, therapeutic relationship); (3) metaphors (i.e., use of cultural specific idioms or symbolic expressions); (4) content (i.e., use of cultural knowledge about values, customs, and traditions shared by specific ethnocultural groups); (5) concepts (i.e., use of culturally informed concepts of illness); (6) goals (i.e., formulating culturally informed and consensual treatment goals); (7) methods (i.e., following culturally informed procedures to achieve treatment goals); and (8) context (considering patient’s social, economic, and political context including immigration history, level of acculturation, developmental stages, availability of social support) also see [38]. However, considering that, for example, over 30% of people in the 2021 Canadian census identified as belonging to PDEGs, with more than 450 diverse ethnocultural (including Caucasian subcultures) and 450 linguistic origins [59], it would be extremely challenging, if not impossible, to culturally adapt each of the recommended elements of psychological intervention in a multicultural setting [60]. Such an undertaking would require making stereotypical assumptions and sweeping generalisations about PDEGs which contradict the very purpose of cultural adaptation of psychological interventions. Further, there are concerns around fit vs. fidelity in comprehensive cultural adaptation [37,54], especially around what elements of the intervention should be culturally adapted, as we do not fully understand the “key ingredients” and mechanisms of change in psychological interventions [55,57,61]. Furthermore, as comprehensive cultural adaptation involves investing significant resources and the fact that general methods of interventions work fine in most contexts [62], it could be argued that cultural adaptation should be initiated only when there is evidence that the original intervention does not have a cultural fit with the target population or loses effectiveness due to a lack of cultural fit [63]. Nonetheless, in light of the growing ethnocultural diversity in the Canadian population and the consequent diversity in cultural concepts of distress, efforts are required to better understand and address challenges faced by PDEGs in order to reduce the current high unmet mental health care needs of Canadians and ensure culturally informed mental health care for PDEGs. 

### 1.2. Patient-Oriented Approach

There is a gap in research on how to make improvements to ICBT in routine care, where individuals come from diverse ethnocultural backgrounds and comprehensive cultural adaptation to all possible groups is not realistic. One potential approach that is generally recommended in Canada to make improvements to healthcare is a patient-oriented research approach [64,65]. Patient-oriented research advocates for patient (Patients here refers to people with lived experiences as well as informal caregivers, including family and friends [63]) involvement as partners in all areas (e.g., design, conduct, evaluation, and dissemination) of health research to ensure that patients receive the right treatment in the right place at the right time [64,65]. This approach focusses on patient-identified priorities and outcomes, a multidisciplinary team approach, and aims to integrate knowledge into practice [64,65]. As such, a patient-oriented approach can be used to adapt ICBT to improve access and engagement in ICBT by PDEGs and may provide an alternative to cultural adaptation as suggestions on what and how to adapt emanate directly from the patients and other representatives. In this approach, PDEGs can assess the content and delivery process of ICBT and then indicate what elements need to be tailored to make it accessible and engaging for individuals of their ethnocultural background. Given that patient-oriented research emphasizes reflecting on what matters to patients the most, this approach does not impose disciplinary thinking or preconceived assumptions about PDEGs during the adaptation process. 

Although there are no clear guidelines yet on how to conduct and engage patients and community representatives in patient-oriented research(see [66]) and, by implication, in adapting psychological interventions, engaging PDEGs in health research and clinical practice can be expected to generate diverse perspectives and knowledge to guide improvements in healthcare services [67,68,69,70,71].

### 1.3. Objectives

The purpose of the current paper is to describe a patient-oriented research process to improve ICBT for use within a routine online mental health clinic that serves individuals from diverse ethnocultural backgrounds. In this study, we formed a multidisciplinary working group made up of diverse representatives and specifically sought to answer the following question: “How can ICBT be improved for PDEGs within a routine online mental health clinic?” The study was conducted in three phases: (1) archival qualitative data analysis of written feedback on ICBT from past users; (2) interviews with current patients, community members, and therapists to identify perceptions of ICBT; and (3) iterative ICBT improvements based on phases 1 and 2 (see Figure 1). 

## 2. Materials and Methods

### 2.1. Setting 

The current research was conducted within the Online Therapy Unit (OTU), which is an online mental health clinic based at the University of Regina but funded by the Government of Saskatchewan to deliver ICBT to residents from across Saskatchewan on a routine basis. According to the 2021 Canadian census, Canada has a rich ethnocultural diversity with over 450 ethnic and 450 linguistic origins, 200 places of birth, and 100 religions, as reported by the Canadian population in 2021. While ~70% of the 38.25 million Canadians identify themselves as Euro-Canadian Caucasian, 6.1% are Indigenous peoples (e.g., First Nations, Métis, and Inuit), and ~25% self-identify as belonging to other diverse ethnocultural groups [59]. More than 1.13 million people have made Saskatchewan their home. Saskatchewan’s population is primarily Euro-Canadian Caucasian; however, 17.0% self-identify as Indigenous peoples, and 14.4% self-identify as from other diverse ethnocultural groups [59]. A recent analysis of ICBT utilization trends over six years (i.e., 2013–2019), see [72], shows that there has been consistently lower (~10%) participation from the PDEGs in the ICBT Course. Research ethics approval for this study was obtained from the ethics board of the University of Regina, and all the research participants signed an informed consent form before study participation.

### 2.2. Wellbeing Course

The main ICBT program offered by the Online Therapy Unit is the Wellbeing Course (the Course here forward), originally developed at Macquarie University in Sydney, Australia [73] and implemented and researched broadly within Australia. The Course is transdiagnostic and designed to address both depression and anxiety symptoms. To access the course, patients begin by completing an online screening questionnaire, followed by a telephone interview. During this interview, ICBT service providers determine if patients meet basic eligibility for the Course (i.e., ≥18 years of age, Saskatchewan resident, endorse depression and/or anxiety, have access to a computer and the Internet, are able to provide a healthcare provider emergency contact), or are experiencing concerns outside of the scope of the Course (i.e., are at a high risk for suicide, have significant alcohol and/or drug usage, or are experiencing unmanaged psychosis or mania). Patients who do not meet the eligibility criteria are directed to more appropriate local services. The Course includes five online psychoeducational lessons addressing (a) an understanding of the cognitive-behavioural model and identifying the symptom cycle; (b) thought monitoring and challenging; (c) de-arousal strategies and pleasant activity planning; (d) graded exposure and behavioural activation; and (e) relapse prevention. Each lesson consists of psychoeducational slides, additional downloadable readings, and activities (i.e., do-it-yourself (DIY) guides, frequently asked questions, and case stories). Patients can also review additional resources addressing a wide range of topics to help them deal with problems not covered in the five core lessons. The Course is completed over ~8 weeks with once-weekly therapist support offered via secure email message or, on occasion, via telephone. Therapists review patients’ progress through weekly symptom measures completed before each lesson, answer patients’ questions about the course, and provide guidance on how they can develop their knowledge and skills gained through the course material. Weekly automated email reminders are also sent to support patients’ continued use of the Course. 

The Online Therapy Unit team has conducted several research studies exploring different aspects of the Course, including outcomes under diverse therapeutic conditions, e.g., [74,75,76,77,78,79], utilization of the Course over the years, e.g., [72], and patient perspectives on the strengths and challenges of the Course [80,81]. However, none of these studies were specific to PDEGs [82].

### 2.3. Working Group

Following the Canadian Strategy for Patient-Oriented Research (SPOR) guidelines [64,65], a working group consisting of a multidisciplinary (clinical psychology, social and cultural psychiatry, public health and administration, social work, counselling psychology, community service, etc.) team of 11 members was formed (titled Patient-Oriented Research in Diversity Impact—Working Group) to conduct the current research and the adaptations of the Course. These members included a person with lived experience (PWLE; n = 1), representatives from community-based organisations (CBOs; n = 4), an ICBT manager (n = 1), ICBT therapists (n = 2), and researchers (n = 4). In this working group, the PWLE, all the CBO members, and two of the researchers (including a trainee) self-identified as PDEGs. The working group meetings were called through email and scheduled based on member availability. Meetings were held over Zoom, with agenda packages circulated prior to the meeting and minutes circulated afterwards. The working group formally met five times throughout the project period (February 2022 to February 2023), but feedback was also provided via email or smaller Zoom meetings during the project. Community members and PWLE each received an honorarium in the amount of $50 after each meeting. 

### 2.4. Phase 1: Archival Data Analysis

Written feedback was extracted from 261 former patients who self-identified as PDEGs or mixed ethnicities and participated in the course between November 2013 and December 2020. An analysis was completed on patient responses to four open-ended questions asked 8 weeks after patients enrolled in the Course about what was most helpful, their likes, dislikes, and what could be improved within the Course. Thematic analysis [83] was used to examine patients’ written feedback. The archival data was coded and categorised by a research trainee (with a bachelor’s degree in social work), a PWLE (with a bachelor’s degree in psychology), and a research associate (AW) after the coders were trained on qualitative data coding as well as on using NVivo for coding and categorizing the qualitative data. 

### 2.5. Phase 2: Semi-Structured Interviews with Patients, Community Representatives, and Therapists 

Patients. Patients who self-identified as being PDEG were asked during the screening process if they would consider participating in an interview to share their experiences with the Course. Consenting patients were provided with more detailed information about the interview, a consent form, and were asked via email to schedule their interviews after being enrolled in the Course for 6 weeks. The patient interview included 12 open-ended questions that asked about the patient’s experience with the Course, including likes, dislikes, expectations, cultural relevance of different aspects of the course, perception of therapists’ support, and ways to improve accessibility and utilization of the course for PDEGs. 

Community representatives. To recruit community representatives for the interview, invitations with information about the OTU, the study, a consent form, and the available compensation of a $50 gift card for participating in the interview were sent via email to various CBOs and specific professionals in Saskatchewan. Organizations and professionals were identified through word of mouth as well as website searches. To be included in the study, CBO representatives had to reside in Saskatchewan, Canada, and have personal or professional experience in providing services to PDEGs. These ~20-min semi-structured interviews were completed by a researcher (EV) via telephone and were audio recorded digitally after obtaining informed consent. The interview contained nine open-ended questions focusing on their experiences providing services to PDEGs, including their understanding of how PDEGs manage their mental health concerns and seek help, key barriers to accessing and utilizing existing mental health care, barriers to accessing and using online mental health services, and potential ways to improve the accessibility and utilization of the Course among PDEGs. 

Clinicians. Written feedback was collected from 3 clinicians who provide therapist support in the OTU. Interview questions were sent to the clinicians via email. Clinicians recorded their responses separately and then compiled their responses into one document, capturing their insights in an anonymized written summary of their responses. Therapists responded to four open-ended questions asking them to reflect on their experiences (e.g., positive experience, challenges) working with PDEGs, as well as suggestions to improve the Course and their ability to work with PDEGs. 

Data Analysis. The digitally recorded interviews were transcribed and naturalized verbatim [84,85], and coded and organised according to pre-determined themes (e.g., likes, dislikes, most helpful skills, cultural relevance, key barriers or challenges, ways to improve, etc.) for thematic analysis. The frequency of each code was also recorded for descriptive analysis. The semi-structured interview data were coded by a researcher (EV) and research associate (AW) using NVivo (version 12 Plus) software 

### 2.6. Phase 3: Iterative Course Adaptation Phase

The Course adaptation was completed iteratively and collaboratively by the working group in the following steps: (1) review of findings and development of recommendations based on data; and (2) review of modifications to treatment materials and the process of navigating and accessing material. This process began by presenting the working group with the results of phases 1 and 2, followed by a discussion of conclusions and recommendations. Immediately afterwards, the OTU research team worked on the proposed adaptations. The adaptations were then sent to the working group for review, and any additional feedback was considered.

## 3. Results

### 3.1. Archival Data Analysis

The patient participants in this study represent various ethnocultural backgrounds, including Indigenous (n = 129; 49.4%), Asian (n = 55; 21.1%), Black (14; 5.4%), Hispanic/Latino American (n = 15; 5.7%), Pacific Islander (n = 8; 3.1%), and others (n = 40; 15.3%). The average age of the participants was 34.59 (standard deviation = 11.11, ages ranging from 18 to 72 years), and the majority (n = 178; 68.2%) were women. The majority of participants were educated at a college level or higher (n = 210; 80.45%), married or in a common-law relationship (n = 149; 57.10%), and lived in urban areas with more than 20,000 people (n = 189; 72.41%). A majority of the patients (n = 186; 71.26%) learned about the Course from health professionals, 38 (14.56%) were referred by family, friends, and community organisations, 31 (11.88%) found the Course through media (online, print, and other), and 6 (2.30%) from other sources. The analysis of archival feedback from former patients revealed that PDEGs found the core techniques and skills in the Course helpful in managing their symptoms of anxiety and depression. Out of 261 PDEGs who provided feedback, 165 (63.22%) described Thought Challenging as the most helpful skill they learned in the Course. Patients attributed Thought Challenging to gaining a healthy mindset and managing their negative thoughts. The second most helpful skill reported was Controlled Breathing (27.20%), which patients found to improve their daily functioning. Likewise, Graded Exposure (n = 71; 17.62%), Activity Planning (n = 46; 16.86%), and Cycle of Symptoms (n = 22; 8.43%) were among the other techniques that patients found helpful in dealing with their concerns, while only 8 patients (3.06%) reported not learning any new skills (see Table 1 for the details). 

When asked what the patients liked the most about the Course, the majority said they liked the DIY Guides (n = 112; 42.91%), which are worksheets with activities to help patients practice the skills they have learned in the corresponding lessons. Patients reported that the DIY Guides were difficult to practice but beneficial when utilized and served as a helpful summary of the corresponding lessons. Following the DIY Guides, 88 (33.72%) patients appreciated the Stories and their ability to make them feel less alone. Moreover, many patients reported liking the Additional Resources (n = 55; 21.07%), accessibility (n = 44; 16.86%), therapist support (n = 39; 14.94%), the overall Lessons (n = 25; 9.58%), and the educational approach to symptom explanation (n = 9; 3.45%) (see Table 2 for the details). 

Almost all patients indicated that they liked the Course overall and recognized its contribution to their symptom management; however, areas for improvement emerged when asked what they disliked about the course. While 84 (32.18%) patients provided only positive responses, stating that they did not have any dislikes about the Course, 113 (39.46%) and 53 (20.31%) reported some issues with the content and the process, respectively. Most notably, an area for content improvement was the Stories, with 34 (13.03%) indicating that some Stories were difficult to relate to in terms of demographics, the issues faced, and the experiences described by patients in the Stories, while others reported the Stories were too lengthy. The second area of improvement came from dislike of the website interface or utilities (n = 33; 12.64%). Some responses included a dislike for the amount of reading per Lesson, and other responses indicated a desire for audiovisual elements to accompany the readings. Thirty (11.49%) patients did not provide a response (indicated by N/A as their response) to the dislike question (see Table 3 for the details).

### 3.2. Interviews

Patients. The patient interviews (n = 16) conducted with active participants in the Course confirmed the findings in the archival data analysis. Whilst one of the interviewed patients had no expectations and two patients felt that the Course did not meet their expectations, the majority (n = 13, 81.25%) reported a positive experience working on the Course, explaining that the course met or exceeded their expectations. They shared that the core content of the Course, such as the skills and exercises, was beneficial for managing symptoms of anxiety and/or depression. Additionally, 9 (56%) patients also reported that the Stories were relatable and helped them to feel “less alone” as well as to recognize similar behavioural patterns in their own lives. For example, one 22-year-old Indigenous patient shared her experience as stated: “I liked all, like the Stories and stuff- that it’s very relatable, and like having different perspectives on it too”. 

The therapist support was an appreciated aspect of the course for those who utilized it, with 13 (81.25%) patients reporting having a positive experience. The following is an excerpt from an interview with a patient: 

“…She [therapist] was very dedicated and she was very capable. Even a little hint, she catches it and she starts guiding me. Also, she gave me a brief idea about the Course and how it applies to me in particular… she was very helpful”.[42, Male, Southeast Asian]

When asked about the relevance of the Course materials in relation to their ethnocultural background, half of the patients (n = 8) reported that the Course fit with their cultural beliefs and values. Some mentioned that because the Course is generic and does not adhere to one particular culture, it is appropriate and accessible to a variety of cultures, which is evident in the following two representative interview excerpts:i.“It was all good for me. I think it fit with me perfectly. I think that it doesn’t really matter if I’m like Asian…I think it was accessible for everyone”. [34, Female, Asian]ii.“Yes. There wasn’t really anything cultural about it, right? I don’t think anyway”. [22, Female, Indigenous]

One patient reported that the Course did not align with their cultural beliefs, and this is because their culture does not support seeking treatment for mental health problems. This patient’s culture believes that mental health help should be sought within the family. The remaining patients reported that nothing stood out for them in terms of cultural alignment (n = 5, 31%) or did not provide a response (n = 2, 12%). Nevertheless, these interviews also revealed elements of the Course that patients did not like; 6 (37.5%) patients did not find the Stories relatable or helpful. One patient noted, however, that this is not necessarily because of ethnicity but because of the type of mental health issues described in the Stories. Another patient remarked that although they were happy to see diverse ethnicities represented through imagery, the two consistent character vignettes (included in the Course as an example to demonstrate how the skills and techniques can be applied) were Caucasian/White. This may have contributed to the 2 (12.5%) patients reporting that there is a lack of diversity in the Stories characters or examples, as one patient discusses in the following interview excerpt: 

“There is nothing about being a newcomer… it was one of the factors that gave me a lot of anxiety, I think. Having this shock that I’m in a new country with no connections with other people. I think that’s one [reason for anxiety]. I think it [the Course] was not made with a newcomer in mind… it’s more of a general”.[40, Male, Black]

When asked for suggestions for specific improvements to make the Course better for people from their ethnocultural background, some patients (n = 8, 50%) indicated the need for diversifying case stories with examples describing challenges unique to PDEGs. The remaining patients suggested accommodations for the hearing impaired (n = 1, 6.25%), improving website interface functions (n = 2, 12.5%), more time with the Course (n = 1, 6.25%), or provided no suggestions (n = 2, 12.5%). Patients also had the opportunity to share any recommendations that would have improved their experience with the Course that had not been shared yet, which included a larger font (n = 1, 6.25%), increasing therapist support (n = 2, 12.5%), providing more time with the lessons (n = 2, 12.5%), summarizing and shortening the reading in the lessons (n = 1, 6.25%), and audio elements (n = 2, 12.5%), while the remaining patients did not suggest improvements (n = 8, 50%). Evidently, the patients’ suggestions for improvement focused on the general accessibility of materials, with diverse suggestions provided and no particular suggestions to improve the key elements (e.g., cognitive-behavioural techniques) of the Course.

Community representatives. Six out of seven (85.71%) community representatives reported language and stigma as the most common barriers to seeking and receiving mental health services by PDEGs. They explained that due to the fear of experiencing stigma (e.g., being labelled as “crazy”), which is rooted in patients’ cultural knowledge, beliefs, and practices related to mental health and illness in their countries of origin, PDEGs are reluctant to seek help and to share their [mental health] problems. 

Further, the community representatives highlighted that PDEGs regard and deal with mental health concerns differently than the mainstream health care system in Canada. They noted that in some cultures, discussing mental and emotional wellness is considered taboo, so people may be averse to talking and identify themselves as experiencing mental illness symptoms, and thus not seeking help at all. They regard their problems as emanating from precarious life circumstances (e.g., attribute low mood and irritability to not being able to find a proper job), or as inflicted by external forces (e.g., “Satan is making me tired”), so they prioritize finding a job over dealing with low mood and irritability. In addition, they may believe that they can manage their problems themselves or that “time will heal things”. Consequently, PDEGs often seek help from informal health care sectors such as community support systems (e.g., traditional healing, religious organizations) and find solace in gatherings with community such as “drumming circle”, see [86], spiritual ways such as prayer, and natural ways such as crafts, music, and art, and in speaking to their friends, religious leaders, and people they feel safe with. Relatedly, some community representatives opined that due to cultural differences, PDEGs may not have the adequate language to express their mental health concerns in a way that is comprehensible to service providers and may not be knowledgeable about mainstream Canadian ways of understanding mental health problems and finding resources that can help. 

Community representatives also reiterated some putative aspects of the Course that could act as barriers to PDEG using the course, such as having access to the internet and devices, English language and computer literacy, and a distrust that confidentiality can be maintained in services provided online.

When asked what might help PDEGs in access and use the Course, a majority (n = 5; 71.43%) of the community representatives suggested offering services in the patient’s language. Additionally, increasing awareness of ICBT, providing help with and access to the required technology, including motivational counselling sessions, educating on mental health by including psychoeducational materials on the website and through presentations in community organisations, simplifying and using destigmatizing language, offering therapists from different cultural backgrounds, and advertising patient testimonies from diverse backgrounds were among other suggestions offered by the community representatives. 

Therapist feedback. The therapists shared that the patients who had participated in the Course and had received their support voiced their gratitude for the availability of services for people in rural communities. The patients acknowledged that the Course had provided them with an opportunity to share their emotional and mental health problems, which they otherwise would not have felt comfortable addressing within their cultural group. Concurrently, the Course content validated patients experiences with mental health problems as it helped normalize their mental health concerns. Further, patients reported the skills learned were new to them and something they wished they had learned earlier. Therapists noted that the learning was mutual in that both the patients and the therapists learned more about cultural traditions from each other. 

The therapists’ indicated their ICBT training materials lacked attention to some cultural issues. As an example, therapists noted that they have found it difficult to help navigate unfamiliar stresses such as those experienced by individuals and families who have lived in war-torn countries. Examples of this would be intergenerational family dynamics and expectations of success from relocating to Canada. Therapists shared that some patients commented on the lack of cultural information within the course, for example, a list of helpful activities appears in a lesson but lacked cultural examples such as ‘beading’. The therapists also reported that sometimes they struggled to understand patients’ communication, especially when patients were newcomers with limited English language competency, preventing patients from writing detailed messages and also having difficulty conversing by phone. 

To improve therapists’ own abilities to better support PDEGs, therapists suggested including examples of culturally diverse situations, skills, and techniques in their training materials. Furthermore, they suggested that the Course incorporate more culturally relevant Stories and an additional resource to help individuals and families understand mental health issues. They advised that broadening representation in the Course images, advertisements, and social media would improve course awareness, and collaborating with PDEGs to review the Course materials would ensure inclusivity. 

### 3.3. Improvements

The Course modifications were performed iteratively by the working group as follows: (i)First, the five lessons in the Course were thoroughly reviewed for opportunities to simplify language by the OTU researchers and clinicians. This step involved identifying English language and Euro-American culture-specific idioms, metaphors, examples of activities, and academic vocabulary. These phrases were replaced with more general and commonly used language to ensure the Course was accessible to everyone, regardless of education level or cultural understanding.(ii)Second, the Course content was reviewed by PWLE, trainees, and CBO members to identify any issues with case stories, DIY activities, language, cultural relevance, and techniques described in the materials. At this stage, the aesthetics and user-friendliness of the Course materials were also reviewed. The group provided suggestions for improvement.(iii)Third, the final adaptation of the Course was informed by the recommendations from the working group and the findings from phases 1 and 2.

Drawing on the findings above, the working group did not identify the need to make major changes to the Course but suggested some changes to the materials to improve inclusivity. Furthermore, they made suggestions for improving the training of therapists and some therapist practices, as well as outreach practices, which are elaborated on below.

#### 3.3.1. Course Materials

Language. In all lessons, English idioms were removed to prevent PDEG’s alienation from understanding. For example, in a Lesson 1 vignette, a character referred to their physical symptoms of anxiety as being “wound up”. This was changed to “feeling nervous” to eliminate misunderstanding. Another example was found in Lesson 2: a statement was changed from “don’t just put up with it” to “don’t just accept it”. While these are common phrases used in some cultures, these expressions can exclude some individuals, which impacts their ability to gain all the benefits the Course offers. Along with removing idioms, more specific and descriptive wording was implemented to improve understanding. Removing idioms and making minor wording changes were made to the DIY Guides as well to improve clarity. One DIY Guide was altered to include examples of activities to help manage mental health problems, such as attending cultural celebrations, beading, going to a place of worship, and Tai Chi. 

Imagery. Stock imagery is used throughout the Course and was reviewed to ensure representation of diverse ethnicities. Many images were changed to include more PDEGs with visible differences in abilities, gender, and cultural expressions. This included changing the image depicting a reoccurring character that appears in the Course. The character formerly was a young Caucasian female who accompanied a young Caucasian male and was changed to an image of a middle-aged Asian female. 

Audiovisual content. To address the patient’s aversion to the amount of reading throughout the Lessons and their request for audiovisual elements, videos were embedded on the website at the beginning of each of the five Lessons. Each video is approximately 3 min and serves as an introduction to the Lesson patients are about to begin. 

Acknowledgement of culture. Lesson content was updated to incorporate diverse situations and thoughts and acknowledge cultural differences. During the Course introduction, presentation slides were created and dedicated to recognizing culture and emotional wellbeing. Elements of culture are explained, and it is acknowledged that culture impacts how we interact with and understand the world. It is also explained that culture shapes our understanding and communication of emotional wellbeing. The slides ask patients to reflect on their culture and its influence on their beliefs around distress, perceptions of their abilities to alter challenging situations, and ways of seeking treatment. The idea that people in Canada from all social and cultural backgrounds experience anxiety and depression was included, as well as examples of social challenges that people in Canada may encounter, such as discrimination or financial insecurity. The second Lesson included another mention of the complex social, cultural, political, and economic differences that influence perceptions. Social stigma and cultural differences in knowledge about mental health problems were named as reasons that prevent fully understanding symptoms. Stigma is mentioned again as an explanation for why some may like confidential ICBT. 

Stories adaptation and creation. All Stories and examples were examined to incorporate feedback from phases 1 and 2. Minor wording changes were made to allow the Stories to be more specific and concise. In an attempt to shorten the Stories, ideas and explanations were reworded to be more succinct. One of the Stories was formerly about a Caucasian male, and the image was changed to depict an Indigenous male. Indigenous cultural elements were added to the narrative as well, such as struggling with transitioning from living on a reserve to an urban centre. The names of two characters were changed to reflect someone of East Asian descent and someone of Hispanic descent. In one of the Stories, a character’s main issues were changed in an effort to present differing circumstances. In this instance, the character was a victim of domestic violence and struggling in their relationships. To address the need for better representation, three new Stories were created and added to the Course. The first is a narrative about a 74-year-old Caribbean woman who immigrated to Canada when she was younger and is now struggling with the loss of her husband and aging. The second explores the challenges of immigrating to Canada due to war while experiencing loss, culture shock, and financial hardships as a 40-year-old mother from Lebanon. Lastly, an international student’s experience of coming to Canada from China and dealing with the social implications, finances, family pressures, and language proficiency was added as a story. 

#### 3.3.2. Training

ICBT service providers at the OTU (n = 9, 8 Social Workers, 1 Certified Counsellor) attended training conducted by a researcher (RS). This training included explaining cultural concepts of distress (CCD) and the importance of understanding CCD in assessments and providing culturally informed care. This was achieved by reviewing the DSM-5 entry on CCD and providing examples of ways in which different cultures understand and express mental health problems through cultural idioms of distress, cultural explanations, and cultural syndromes [30]. Teachings were also completed on becoming culturally informed and how to apply this to routine practice in ICBT therapist support. To do this, culturally informed assessment guides were reviewed, such as the Cultural Formulation Interview [30], The Explanatory Model Interview Catalogue [87], and the McGill Illness Narrative Interview [88]. This didactic teaching involved audiovisuals, group discussions, and role-play techniques to facilitate learning. A second training session was conducted to provide ICBT service providers with a summary of the findings at the midpoint of the research study. This session included presenting the research findings and themes from phases 1 and 2. Clinicians at the OTU were also able to attend a Cultural Awareness Presentation hosted by a non-profit organization that provides integration and settlement services for newcomers to Canada. Lastly, the ICBT service providers were educated on the iterative course adaptations that lead to updating therapist protocol guides. The protocol guides summarized the Course changes and provided email templates that included prompts to elicit culturally informed conversation and reflection by patients. 

#### 3.3.3. Outreach

One important issue that surfaced in discussions with community representatives as well as working group members is the lack of awareness the general public has about the OTU. Thus, plans were made to improve outreach, which is now underway and seen as an ongoing process to improve knowledge of the OTU among PDEGs. In order to increase recruitment and diversify the intake of the Course, outreach techniques have been refined. A video was created by researchers that summarizes OTU services and highlights the updates to the Course through this study. This video has been shared online and with community organizations. A presentation was developed and is currently being offered to community organizations who are interested in learning more about OTU services, the process and findings of this study, and the incorporated changes to improve ICBT for PDEGs. These are new additions to our regular recruitment practices, which include mailing promotional material to medical clinics and community organizations as well as offering presentations to service providers. 

## 4. Discussion

We applied a patient-oriented research approach in an effort to improve ICBT for PDEGs in a multicultural routine mental health care setting. Archival data analysis and interviews were conducted to build our knowledge on what aspects of the Course and practices are valuable and what elements need modification to improve the Course for PDEGs. The results show that almost all PDEG patients liked the Course overall and recognized its contribution to their mental health. They also suggested improvements, such as offering services in the patient’s language, including therapists from diverse ethnocultural backgrounds, incorporating audiovisual materials to reduce the amount of required reading, adding materials on social stigma, diversifying the imagery, case stories, and examples of activities practiced in different cultures in the Course to make the Course more relatable to PDEGs. We were able to implement most of the recommended modifications, with the exception of changing the English language of the Course and adding therapists from PDEGs. Although we have simplified the language, educated our therapists on understanding cultural diversity and delivering culturally informed care, and are exploring the possibility of adding an online translation option to the Course, providing PDEG services in their own language with an ethnoculturally matched therapist is not feasible as we may potentially receive patients from over 450 ethnocultural backgrounds and languages [59]. 

Importantly, most of the suggested modifications to the Course were primarily on peripheral elements [49,54], such as the aesthetics of the web-based interface, stories and examples, and the process, with no major suggestions on the cognitive-behavioural model and techniques. In fact, cognitive-behavioural techniques such as thought challenging, controlled breathing, and graded exposure were among the most liked and utilized elements of the Course (see Table 1). This implies that the core cognitive-behavioural constructs and techniques used in the Course were relevant and acceptable to PDEGs. As quoted in the results, some community representatives explicitly noted that the skills and techniques presented in the Course were general and not specific to any one culture. Further, as the core cognitive-behavioural elements of the Course remained largely unchanged, we have no concerns about fidelity to the original model of the treatment, and we can expect similar effectiveness from the adapted treatment. Similar findings were obtained in our previous studies eliciting patient perspectives on the strengths and challenges of the Course [81,89]. These findings provide support for our use of a patient-oriented approach instead of a cultural adaptation model in that the adaptation of psychological interventions should be tailored to the current context-based perceived needs of patients. Having said this, we are aware and do not intend to purport that cognitive-behavioural techniques are acultural [90] nor to devalue the importance of understanding cultural influences on all aspects of mental health and illness. The need to culturally adapt psychological interventions when used outside of the cultural context where they were originally developed is still of great value to patients. However, comprehensive cultural adaptation may not always be possible, especially in routine care settings where patients of diverse ethnocultural backgrounds are provided services. Therefore, with this study, we explored an alternative avenue to identify and improve ICBT for PDEGs when comprehensive cultural adaptation is not feasible.

Overall, our research revealed that a patient-oriented approach to adapting psychological interventions is practical. Since those involved in treatment identify and decide which elements of the intervention are adapted and how adaptation should take place, it does not rely on stereotypical assumptions about individual culture and its influence on mental health and wellbeing. A patient-oriented approach is informed by patient needs and preferences without major concerns about compromising the fidelity of the evidence-based intervention. 

### Strengths, Limitations, and Future Directions

Our working group was composed of a diverse range of ethnocultural backgrounds, disciplines, and lifestyles. Through archival data analysis and interviews, we sought to gain insight into the issues that mattered most to our patients. Despite approximately 43 to 50% of Indigenous peoples participating in our research, we had no representation of Indigenous peoples on our working group. As such, this may have prevented us from properly recognizing and responding to the issues specific to Indigenous cultures in the Course. Further, most (four out of seven) of the community representatives interviewed were involved in providing services primarily to newcomers. Therefore, the experiences they shared of working with PDEGs may not reflect a comprehensive understanding of the PDEGs who have lived in Canada for longer periods, as the level of acculturation as well as social and contextual factors play an important role in this regard. To rectify these limitations, future work should involve collaborating with Indigenous peoples and involve more community representatives who have worked with PDEGs in various social contexts. 

Furthermore, although it has not been discussed in this article previously, it is important to mention that the requirement for a Saskatchewan-based medical doctor to be listed as an emergency contact may act as a barrier to accessing our services, particularly for newcomers and PDEGs. Indeed, approximately 20% of Canadians lack both a family doctor and a regular healthcare provider [91,92]. Perhaps, due to this requirement, we are unintentionally channelling our patients via health professionals, where they obtain psychotropic prescriptions before coming to us. This is something that we need to reconsider to improve the accessibility of our services to newcomers and PDEGs.

The interviews with community representatives and patients revealed one clear barrier to the Course is English language proficiency, and as such, the working group and community representatives recommended offering translations of the Course to enhance ICBT for PDEGs. Resources are not currently available to follow up on this recommendation, but in the future, we hope that it will be feasible to add translation software to the website and study the usage and impact of this on service outcomes among PDEGs.

Increasing the intake of PDEGs in the Course will require further changes to recruitment methods. As we continue to offer presentations on OTU services, we will extend our outreach and build relationships with organizations that specialize in newcomer and settlement services. In addition, we plan to improve our promotional material to include more diverse imagery. Stigma has been identified as a significant barrier, which enforces the importance of representation in advertising as a means to normalize PDEGs seeking treatment. This could also be performed by explicitly including demographic information and service outcomes in advertisements. The promotional material could be further improved through translations into different languages and visibility in the corresponding communities.

Another identified barrier that could not be addressed due to limited resources was accessibility to the internet and the required devices necessary to access the Course. Our objective is to further develop relationships with CBOs and locate organizations that offer internet access to work in partnership to refer patients to our services and vice versa.

Future research is now required to assess the outcomes and feedback on the newly adapted Course to verify iterative changes have benefitted engagement, satisfaction, and outcomes. Furthermore, it would be beneficial to interview patients shortly after beginning the Course in order to capture feedback from patients who may not remain in the course for 6 weeks. Longer term, it would also be valuable to study the impact of improved outreach on the use of the Course by PDEGs.

## 5. Conclusions

This paper elucidates how patient-oriented research can be used to identify and make improvements to ICBT. The suggestions provided may prove beneficial to other groups in a similar situation. Although clinical studies are required to determine if the patient-oriented adaptations have improved the clinical effectiveness of the Course, this study shows that patient-oriented adaptation is practical and useful in multicultural routine care settings where comprehensive cultural adaptation of internet-delivered psychological intervention may not be feasible.

## Figures and Tables

**Figure 1 healthcare-11-02135-f001:**
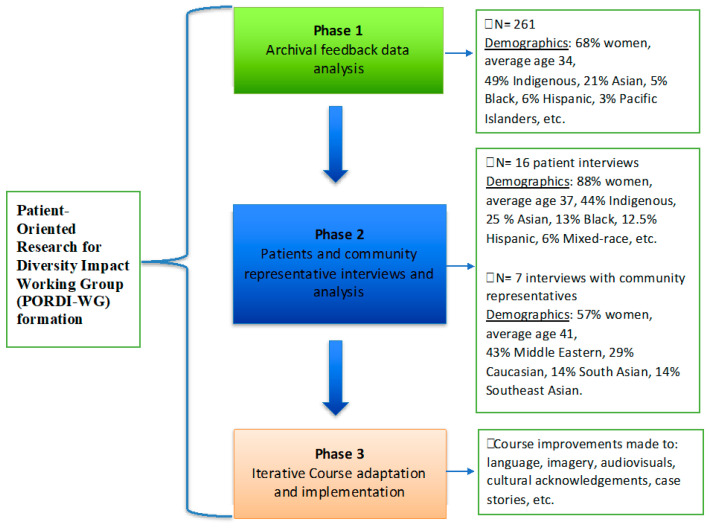
Adaptation process.

**Table 1 healthcare-11-02135-t001:** Patient responses to “For you, what was the most helpful skill taught in this course?” (N = 261).

Most Helpful Skills	Examples	n	%
Thought challenging	“Thought challenging because examining your negative thoughts helps to put them in perspective and to create a healthier view of them”. “Thought challenging was the most helpful skill in this course because without it I would not have been able to overcome my negative mindset. Through it I was able to change my ways of thinking into a more positive one. It was a steppingstone for me to be confident in learning new skills for the other lessons. I was able to be hopeful that I can overcome my symptoms”.	165	63.22
Controlled breathing	“Controlled breathing was my most helpful skill specifically from this course, given how simple the technique is, I think I found it extremely beneficial to complement other strategies I was starting to work on as well”. “controlled breathing has greatly improved my functioning throughout daily tasks”	71	27.20
Graded exposure	“Graded exposure will probably be the most helpful skill in short and long term. It is very helpful for me and will benefit me for years; I am sure of it”. “I feel graded exposure is the most helpful skill in this course, because I used to play games and watch videos to escape from the real world, but now I can plan to do something every day”. “…graded exposure was helpful in getting out and about. I also noticed that I started playing chess again and reading more often. In other words, I am enjoying past activities rather than doing nothing”.	46	17.62
Activity planning	“Becoming more active struck me as the most important skill—I actually noticed a direct impact in my life once I started to apply activity scheduling in my life (as much as I could)”. “Becoming more active has been what gets me to the point of happiness, when I sit too much the symptoms increase drastically and I become so overwhelmed to the point of being numb and I can’t feel anything even love for my loved ones”.	44	16.86
Cycle of symptoms	“I really liked learning about the cycle of symptoms and realizing what healthy versus un-healthy anxiety/stress is. I really thought I was more educated on this topic but had a real eye opening experience when I started to learn about it”. “Increased awareness of symptoms and what can be done before it escalates”.	22	8.43
Overall positive reflection	“All of the lessons were very helpful”. “I can’t choose just one this course was challenging but beneficial in learning that I can help myself”	13	4.98
No new skills learned	“There wasn’t a most helpful skill. I found that I was already doing the things that had been suggested”	8	3.06
No response provided	N/A;	3	1.15
Relapse prevention	“The relapse plan. Even reading the three reasons we relapse are all items I certainly relate to and know that by recognizing any of those I will set myself up for make changes that will be positive going forward”.	3	1.15

**Table 2 healthcare-11-02135-t002:** Patient responses to “What did you like about the course?” (N = 261).

Most Liked Elements	Examples	n	%
DIY Guides	“…even though I knew about most lessons beforehand, the DIY guides helped me remember the lessons and pushed/motivated me to act on what I learned”. “I really liked the DIY guides as they provided great activities as well as they summarized many of the main points of a given weeks lesson. These lesson summaries in the DIY guides where helpful to me as I didn’t always have consistent access to my computer throughout the course, so being able to print off and have the DIY guides available to take places with me provided a good alternative for when I wanted to review some of the most important points of a given weeks lesson”.	112	42.91
Stories	“The stories, I felt a connection to the stories and I could relate to them. Reading how the different skills helped them feel better gave me hope that this actually does work”. “Including examples from other people who are dealing with anxiety and depression helps me realize that I am not the only one who deals with these feelings. It really puts it into perspective that I’m not perfect, but I am a good person and its about time I realized that”. “The stories were nice to read as they made me feel like I wasn’t alone in my symptoms and feelings”	88	33.72
Additional Resources	“I think the resources are helpful that I learn a lot from them”. “…I liked being able to access the supplements at my own pace, I think that allowed me to read the resources for the problems I was facing earlier on which allowed me to start on some strategies sooner rather than later as well, although I eventually went through all the supplements (apart from motherhood I suppose, since being a parent is not part of my life at this time)”.	55	21.07
Accessibility	“Easier access and flexible to accommodate with busy life style”. “How it explains in a simple and logical way the material”. “I liked that I was able to do/complete the course at my own pace and with what worked for me—online”.	44	16.86
Therapist support	“I really liked having someone contact me every Tuesday to let me know they care about my progress. It has helped me make it through this course and not give up”. “I think the setup is excellent, I also believe having someone to check in every week helps to keep the client accountable. I’m pretty sure I would have withdrawn weeks ago if it weren’t for the accountability I felt towards Kerry as she was working so hard to keep me engaged and supporting me in every way she could through email”. “I also really liked emailing with my therapist. It added accountability in addition to giving me feedback, suggestions, etc. It was a bit tricky that she would check in on Fridays because that was the end of the week for most lessons, but it ended up reinforcing the inter-connectedness between the different lessons and I think it helped me integrate everything better”.	39	14.94
Lessons	“Each lesson provided valuable insight. I enjoyed making appointments with friends so I would keep doing it”. “The lessons are pretty easy to understand and read”.	25	9.58
Overall positive response	“I liked all aspects of the course and think it is well laid out”.	18	6.90
Educational approach	“I liked how everything was explained, the symptoms and terminologies”. “I liked the way that the course was approached. It wasn’t a looking down upon approach but a down to earth, ‘this is how things are, how can we fix them?’”	9	3.45
No response provided	“N/A”	3	1.15

**Table 3 healthcare-11-02135-t003:** Patient responses to “What did you not like about the course? What should we do to improve it?” (N = 261).

Most Disliked	Example	n	%
Issue with content		103	39.46
Stories	“The stories I didn’t find to be super engaging, mainly just glossed over them” “The stories were a bit hard to relate to since I felt like Glenn and Jo were so much more successful than I was and were so normal/well adjusted”. “It would be helpful if you could include some scenarios where people are trying to deal with difficult relationships, or people around them who are very negative and toxic. This was the cause of much of my depression and it would be very helpful to read some examples or stories of people dealing with this type of problem”	34	13.03
Website interface or utilities	“I don’t enjoy reading the font selected to write out the information and surveys. It’s difficult for me to read, the words are squished together and it takes extra effort just to make sure I’m seeing the words properly”. “One thing I would suggest is for the questionary layout, every time when I finish the questions I have to remember to click the “Press to submit” button (which by the way it’s not easy to spot and easily to miss it on the left hand side) before going to the next section, if I forgot then I have to restart all over again. This is not user friendly layout and it can be done in a much better way for end-users”. “I guess improve upon the interface for the surveys; you have any idea how infuriating it is to fill out the survey and hit “next” instead of “submit” and all your answers get reset because you didn’t hit “submit” first?”	33	12.64
Content not personal relevant	“Lack of personalization”	10	3.83
Additional resources	“the resources while informative about the issue didn’t always offer many Tips for dealing with it	9	3.45
DIY guide	“DIY I thought it was full of activities and not just a summary of the lesson. I thought it was going to be like a booklet of stuff that will help me feel better and not a bunch of the same information that I read”. “I would suggest creating a daily activity that we can follow in the DIY for example. Journaling daily to keep us more on track in the course. I find that I only read the materials once a week and forget to review until the next week”.	7	2.68
Generic information	“Somewhat generic”	6	2.30
Repetitive material	“Lots of repetitiveness. I understand why though. At times, it was a little much”.	4	1.53
Issues with process		53	20.31
Limited time	“I would like to have more time to do the lessons. Maybe shorter lessons and a longest period (e.g., 3 months)”.	27	10.34
Therapist support	“I would like to have more contact with my therapist. Possibly even twice a week instead of just once a week”.	19	7.28
Lack of structure	“Being on my own pace with no forced communication. Allowed me to not keep up with the work and I wouldn’t reach out to the therapist on my own”.	7	2.68
Overall positive reflection	“There wasn’t anything I didn’t like about the course”.	84	32.18
No response provided	“N/A”	30	11.49

## Data Availability

All the data used in the current study are available from the Online Therapy Unit (www.onlinetherapyuser.ca) upon reasonable request.

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
