# Peer review of "Patient-Oriented Research to Improve Internet-Delivered Cognitive Behavioural Therapy for People of Diverse Ethnocultural Groups in Routine Practice"

_healthcare, 2023, doi:10.3390/healthcare11152135_

Round 1
Reviewer 1 Report
The authors did a good job meeting their objectives and appear to be quite familiar with qualitative research. I didn't find anything that needs to be changed or revised. It was interesting and thought-provoking, especially how an online CBT course can work.
Author Response
- The authors did a good job meeting their objectives and appear to be quite familiar with qualitative research. I didn't find anything that needs to be changed or revised. It was interesting and thought-provoking, especially how an online CBT course can work.
The Authors Response: Thank you for reviewing our work and for your kind words.
Reviewer 2 Report
I enjoyed reading the paper and clearly there is a need for more research on online therapy provisions. Your study certainly seems very valuable.
In the Introduction section, it would be nice to see some reviews of previous studies on online therapy sessions. This would further highlight the need for the current study. Also, for a reader who is not from Canada, it would be helpful to to have some specific information on the ethnographic make of the Canadian population.
In the Methods section, it would be helpful to see a little more detail on the demographics of the participants, perhaps in a graph?
It would also be interesting to see more quotes from the interviews.
Adding some visuals, such as graphs and tables, would make it easier for the reader to understand the methodology section.
In the Discussion & Conclusion section, it would be nice to see a statement on what the research team intends to do next on this topic.
Author Response
- I enjoyed reading the paper and clearly there is a need for more research on online therapy provisions. Your study certainly seems very valuable.
The Authors Response: Thank you.
- In the Introduction section, it would be nice to see some reviews of previous studies on online therapy sessions. This would further highlight the need for the current study.
The Authors Response: There are several reviews and meta-analyses published in this regard. Therefore, we have added the following most recent review of online therapies [ see Page 5]
Hedman‐Lagerlöf E, Carlbring P, Svärdman F, Riper H, Cuijpers P, Andersson G. Therapist‐supported Internet‐based cognitive behaviour therapy yields similar effects as face‐to‐face therapy for psychiatric and somatic disorders: an updated systematic review and meta‐analysis. World Psychiatry. 2023 Jun;22(2):305-14.
- Also, for a reader who is not from Canada, it would be helpful to have some specific information on the ethnographic make of the Canadian population.
The Authors Response: We have added the following brief information on the ethnographic make of the Canadian population in the manuscript [see Page 4].
“According to the 2021 Canadian census, Canada has a rich ethnocultural diversity with over 450 ethnic and 450 linguistic origins, 200 places of birth, and 100 religions as reported by the Canadian population in 2021. While ~70% of the 38.25 million Canadians identify themselves as Euro-Canadian Caucasian, 6.1 % are Indigenous peoples (e.g., First Nations, Métis, and Inuit), and ~25% of self-identify as belonging to other diverse ethnocultural groups.”
- In the Methods section, it would be helpful to see a little more detail on the demographics of the participants, perhaps in a graph?
The Authors Response: We have added a figure (see Figure 1) that visualizes the multiphase adaptation process with brief demographic information of participants in each phase of the study.
- It would also be interesting to see more quotes from the interviews.
The Authors Response: We agree. However, considering the length of the paper and also that all the examples given in Table 1 (see Table 1) are quotes from patients, we have not added other quotes at this time.
- Adding some visuals, such as graphs and tables, would make it easier for the reader to understand the methodology section.
The Authors Response: We have added a figure (see Figure 1) that visualizes the multiphase adaptation process with brief demographic information of participants in each phase of the study.
- In the Discussion & Conclusion section, it would be nice to see a statement on what the research team intends to do next on this topic.
The Authors Response: We have added information in the discussion section [ see Page 17]. The following text is added in the discussion section:
“As such, a mixed methods pilot clinical trial is underway to test the effect of the adapted Course benchmarked against the un-adapted Course in terms of patient engagement, satisfaction, and effectiveness.”
Reviewer 3 Report
“Patient-Oriented Research to Improve Internet-delivered Cognitive Behavioural Therapy for People of Diverse Ethnocultural Groups in Routine Practice”( healthcare-2440816)
This manuscript aimed to explore the patient-oriented research to improve Internet-delivered cognitive behavioural therapy for people of diverse ethnocultural groups in routine practice. Using data from diverse sources, the current investigation elucidated the merits and demerits of the current patient-oriented Internet-delivered cognitive behavioural therapy and proposed solutions to relieve the demerits. Overall, this topic is extremely important considering the fact that more and more immigrants have came to Canada and the current findings holds great practical implications. However, some concerns appeared after reading the whole manuscript.
1. There are some important reviews and projects developed already to track this problem and these articles need to be reviewed and discussed.
Spanhel, K., Balci, S., Feldhahn, F., Bengel, J., Baumeister, H., & Sander, L. B. (2021). Cultural adaptation of internet-and mobile-based interventions for mental disorders: a systematic review. NPJ digital medicine, 4(1), 128.
Willis, H. A., Gonzalez, J. C., Call, C. C., Quezada, D., Scholars for Elevating Equity and Diversity (SEED), & Galán, C. A. (2022). Culturally Responsive Telepsychology & mHealth Interventions for Racial-Ethnic Minoritized Youth: Research Gaps and Future Directions. Journal of Clinical Child & Adolescent Psychology, 51(6), 1053-1069.
Naeem, F., Sajid, S., Naz, S., & Phiri, P. (2023). Culturally adapted CBT–the evolution of psychotherapy adaptation frameworks and evidence. the Cognitive Behaviour Therapist, 16, e10.
Hornstein, S., Zantvoort, K., Lueken, U., Funk, B., & Hilbert, K. (2023). Personalization Strategies in Digital Mental Health Interventions: A Systematic Review and Conceptual Framework for Depressive Symptoms. Frontiers in Digital Health, 5, 84.
Simenec, T. S., Gillespie, S., Hodges, H. R., Ibrahim, S. A., Eckerstorfer, S., JUS Media? Adaptation Team, & Ferguson, G. M. (2022). A novel blueprint storyboarding method using digitization for efficient cultural adaptation of prevention programs to serve diverse youth and communities. Prevention Science, 1-13.
Salamanca-Sanabria, A., Richards, D., Timulak, L., Connell, S., Perilla, M. M., Parra-Villa, Y., & Castro-Camacho, L. (2020). A culturally adapted cognitive behavioral internet-delivered intervention for depressive symptoms: randomized controlled trial. JMIR Mental Health, 7(1), e13392.
Shala, M., Morina, N., Burchert, S., Cerga-Pashoja, A., Knaevelsrud, C., Maercker, A., & Heim, E. (2020). Cultural adaptation of Hap-pas-Hapi, an internet and mobile-based intervention for the treatment of psychological distress among Albanian migrants in Switzerland and Germany. Internet Interventions, 21, 100339.
Böttche, M., Kampisiou, C., Stammel, N., El-Haj-Mohamad, R., Heeke, C., Burchert, S., ... & Knaevelsrud, C. (2021). From formative research to cultural adaptation of a face-to-face and internet-based cognitive-behavioural intervention for Arabic-speaking refugees in Germany. Clinical Psychology in Europe, 3, 1-14.
Xiao, L. D., Ye, M., Zhou, Y., Chang, H. C., Brodaty, H., Ratcliffe, J., ... & Ullah, S. (2022). Cultural adaptation of World Health Organization iSupport for Dementia program for Chinese-Australian caregivers. Dementia, 21(6), 2035-2052.
Shroff, A., Roulston, C., Fassler, J., Dierschke, N. A., Todd, J. S. P., Ríos-Herrera, Á., ... & Schleider, J. L. (2023). A Digital Single-Session Intervention Platform for Youth Mental Health: Cultural Adaptation, Evaluation, and Dissemination. JMIR Mental Health, 10, e43062.
Marwaha, J. S., & Kvedar, J. C. (2021). Cultural adaptation: a framework for addressing an often-overlooked dimension of digital health accessibility. NPJ Digital Medicine, 4(1), 143.
2. There are also some critiques about this topic, such as,
Balci, S., Spanhel, K., Sander, L. B., & Baumeister, H. (2022). Culturally adapting internet-and mobile-based health promotion interventions might not be worth the effort: A systematic review and meta-analysis. NPJ Digital Medicine, 5(1), 34.
which need to be discussed in more details.
Minor editing of English language required.
Author Response
- This manuscript aimed to explore the patient-oriented research to improve Internet-delivered cognitive behavioural therapy for people of diverse ethnocultural groups in routine practice. Using data from diverse sources, the current investigation elucidated the merits and demerits of the current patient-oriented Internet-delivered cognitive behavioural therapy and proposed solutions to relieve the demerits. Overall, this topic is extremely important considering the fact that more and more immigrants have came to Canada and the current findings holds great practical implications. However, some concerns appeared after reading the whole manuscript.
- There are some important reviews and projects developed already to track this problem and these articles need to be reviewed and discussed.
Spanhel, K., Balci, S., Feldhahn, F., Bengel, J., Baumeister, H., & Sander, L. B. (2021).Cultural adaptation of internet-and mobile-based interventions for mental disorders: a systematic review. NPJ digital medicine, 4 (1), 128.
Willis, H. A., Gonzalez, J. C., Call, C. C., Quezada, D., Scholars for Elevating Equity and Diversity (SEED), & Galán, C. A. (2022). Culturally Responsive Telepsychology & mHealth Interventions for Racial-Ethnic Minoritized Youth: Research Gaps and Future Directions. Journal of Clinical Child & Adolescent , 51(6), 1053-1069.
Naeem, F., Sajid, S., Naz, S., & Phiri, P. (2023). Culturally adapted CBT–the evolution of psychotherapy adaptation frameworks and evidence. the Cognitive Behaviour Therapist, 16, e10.
Hornstein, S., Zantvoort, K., Lueken, U., Funk, B., & Hilbert, K. (2023). Personalization Strategies in Digital Mental Health Interventions: A Systematic Review and Conceptual Framework for Depressive Symptoms. Frontiers in Digital Health, 5, 84.
Simenec, T. S., Gillespie, S., Hodges, H. R., Ibrahim, S. A., Eckerstorfer, S., JUS Media? Adaptation Team, & Ferguson, G. M. (2022). A novel blueprint storyboarding method using digitization for efficient cultural adaptation of prevention programs to serve diverse youth and communities. Prevention Science, 1-13.
Salamanca-Sanabria, A., Richards, D., Timulak, L., Connell, S., Perilla, M. M., Parra-Villa,Y., & Castro-Camacho, L. (2020). A culturally adapted cognitive behavioral internet-delivered intervention for depressive symptoms: randomized controlled trial. JMIR Mental Health, 7 (1), e13392.
Shala, M., Morina, N., Burchert, S., Cerga-Pashoja, A., Knaevelsrud, C., Maercker, A., &Heim, E. (2020). Cultural adaptation of Hap-pas-Hapi, an internet and mobile-based intervention for the treatment of psychological distress among Albanian migrants in Switzerland and Germany. Internet Interventions, 21, 100339.
Böttche, M., Kampisiou, C., Stammel, N., El-Haj-Mohamad, R., Heeke, C., Burchert, S., ...& Knaevelsrud, C. (2021). From formative research to cultural adaptation of a face-to-face and internet-based cognitive-behavioural intervention for Arabic-speaking refugees in Germany. Clinical Psychology in Europe, 3, 1-14.
Xiao, L. D., Ye, M., Zhou, Y., Chang, H. C., Brodaty, H., Ratcliffe, J., ... & Ullah, S. (2022).Cultural adaptation of World Health Organization iSupport for Dementia program for Chinese-Australian caregivers. Dementia, 21 (6), 2035-2052.
Shroff, A., Roulston, C., Fassler, J., Dierschke, N. A., Todd, J. S. P., Ríos-Herrera, Á., ... &Schleider, J. L. (2023). A Digital Single-Session Intervention Platform for Youth MentalHealth: Cultural Adaptation, Evaluation, and Dissemination. JMIR Mental Health, 10,e43062.
Marwaha, J. S., & Kvedar, J. C. (2021). Cultural adaptation: a framework for addressing an often-overlooked dimension of digital health accessibility. NPJ Digital Medicine, 4 (1),143.
The Authors Response: Thank you. We have thoroughly reviewed the above papers and given serious consideration to the reviewer comment. After careful consideration, however, we have decided to only incorporate two of the above papers into our paper. While the papers identified by the reviewer are relevant, we feel that incorporating them would detract from the introduction. In our paper, instead we have opted to highlight and focus on the most relevant papers as this is not a review paper.
The following articles are cited in the paper now.
Naeem F, Sajid S, Naz S, Phiri P. Culturally adapted CBT–the evolution of psychotherapy adaptation frameworks and evidence. The Cognitive Behaviour Therapist. 2023;16:e10.
Spanhel K, Balci S, Feldhahn F, Bengel J, Baumeister H, Sander LB. Cultural adaptation of internet-and mobile-based interventions for mental disorders: a systematic review. NPJ digital medicine. 2021 Aug 25;4(1):128.
- Minor editing of English language required.
The Authors Response: We have done a through revision of the paper rectifying language issues.
...
Papers reviewed for relevance:
1) Balci et al, 2022 – is s review paper, which deals with interventions designed to promote general health – physical activities and healthy eating. Although these interventions may have relevance in mental health care, this study does not evaluate these interventions for mental health specifically, thus not directly comparable to our study nor it can be taken as a negative finding for culturally adapted evidence-based psychotherapeutic interventions.
2) Böttche et al, 2021 – more homogenous in-terms of language (single linguistic group – Arabic-speaking refugees in Germany). For example: In Böttche et al, 2021 Cultural concepts of distress: a culturally appropriate explanatory model of symptoms was added; socially accepted terms for expressing symptoms (for eCETA only) and assessing suicidal ideation were adapted. However, the point we are trying to make is if you are working in a multicultural routine care setting, how many cultural concepts of distress can you adapt when different cultures have diverse cultural concepts of distress?
3) Hornstein et al, 2023 – is a review paper that describes personalization strategies used in DMHIs in the extant literature. They excluded from the review those interventions that were culturally adapted – i.e., ethnocultural group-based adaptations. They conclude, “Finally, empirical evidence for personalization was scarce and inconclusive, making further evidence for the benefits of personalization highly needed.” Personalization of interventions to individuals seems further farfetched in the context
4) Spanhel et al, 2021 – is a systematic review of culturally adapted internet and mobile based interventions for mental disorders, which finds no difference in adherence and effectiveness of the adapted to the original interventions and note that no included study conducted a direct comparison.
5) Marwaha & Kvedar, 2021 is an editorial and focusing mainly on a review paper by Spanhel et al (2021).
6) Naeem et al, 2023 provides a general overview of various cultural adaptation frameworks, existing evidence and future directions.
7) Simenec et al, 2023 – using blueprint-storyboarding technique describe a method of cultural adaptation of prevention programs. Only this paper talks about adaptability for multiple cultures and involving targeted cultural groups to identify what needs to be adapted and to make those adaptations. Similar to our method in terms of involving targeted groups.
8) Salamanca-Sanabria et al, 2019 – Cultural adaptation example. Adaptations were made to relatively homogenous Colombian college student population
9) Shala et al, 2020 - Cultural adaptation example. Adaptations were made to relatively homogenous Albanian migrants in Switzerland and Germany.
10) Shroff, et al, 2023 – Adaptation example of digital MH interventions for youth (11-17 years old)]
11) Willis et al, 2022 – based on selective literature review provides recommendations for cultural adaptation for racial-ethnic minoritized youth and their families.
12) Xiao et al, 2022 – Example of cultural adaptation of an intervention to Chinese-Australian caregivers (relatively homogenous population).
Reviewer 4 Report
Study describes a process used to revise ICBT for individuals with diverse backgrounds who were seeking therapeutic services. The process relied largely on the analysis of archival data collected during treatment over a 7 year period, supplemented by small sample interviews of current patients and therapists and community stakeholders. Collected information were reviewed by a task force, and the task force used the data to generate recommendations for the therapy approach.
The paper offer an interesting, data based approach to assess treatment strategies. I think there is some merit in the description of that process. I am a little disappointed that there are really no outcome data. The course was revised. Have the revisions led to better outcomes for patients? It seems like the recommendations changed processes, but we don’t know if these processes changed outcomes. I recognize that the authors acknowledged the need for outcome research, and frankly, that is the paper I’d like to see. Whether it would be a horse race research comparing the old and new versions or simply a comparison of the new approach with a waiting comparison group, I think outcome analysis is essential. Despite that, I am recommending publication because the authors highlight a data based approach that can be used to support administrative and clinical decision making. I teach a graduate course on data based decision making for administrators, and I’ll probably incorporate some of the approach outlined in this paper.
Author Response
- Study describes a process used to revise ICBT for individuals with diverse backgrounds who were seeking therapeutic services. The process relied largely on the analysis of archival data collected during treatment over a 7 year period, supplemented by small sample interviews of current patients and therapists and community stakeholders. Collected information were reviewed by a task force, and the task force used the data to generate recommendations for the therapy approach. The paper offer an interesting, data based approach to assess treatment strategies. I think there is some merit in the description of that process. I am a little disappointed that there are really no outcome data. The course was revised. Have the revisions led to better outcomes for patients? It seems like the recommendations changed processes, but we don’t know if these processes changed outcomes. I recognize that the authors acknowledged the need for outcome research, and frankly, that is the paper I’d like to see. Whether it would be a horse race research comparing the old and new versions or simply a comparison of the new approach with a waiting comparison group, I think outcome analysis is essential. Despite that, I am recommending publication because the authors highlight a data based approach that can be used to support administrative and clinical decision making. I teach a graduate course on data based decision making for administrators, and I’ll probably incorporate some of the approach outlined in this paper.
The Authors Response: Thank you. We really appreciate and are delighted that you are considering to include some aspects of our work in your teaching. Your interest in seeing the results/effect of this adaptation resonates with our interest. However, as you have noted, our aim with this was to describe the process of the adaptation work that was done. As we have mentioned in our response above, a mixed methods pre-post pilot study to test the effectiveness, engagement, and satisfaction with the adapted ICBT course is underway. Data of this pilot study will be benchmarked with the data from previous study that used standard (non-adapted) ICBT course.
Round 2
Reviewer 3 Report
Thanks for the revisions and no further concerns.
Minor editing of English language required.